# Stereochemistry of Simple Molecules inside Nanotubes and Fullerenes: Unusual Behavior of Usual Systems

**DOI:** 10.3390/molecules25102437

**Published:** 2020-05-23

**Authors:** Valerij Kuznetsov

**Affiliations:** 1Ufa State Aviation Technical University, K. Marksa, 12, Ufa 450008, Russia; kuzmaggy@mail.ru; 2Ufa State Petroleum Technological University, Kosmonavtov, 1, Ufa 450062, Russia

**Keywords:** nanotube, fullerene, endocomplex, computer simulation, barrier of internal rotation, enantiomer, cis and trans isomers, conformational equilibrium, nitrogen pyramidal inversion

## Abstract

Over the past three decades, carbon nanotubes and fullerenes have become remarkable objects for starting the implementation of new models and technologies in different branches of science. To a great extent, this is defined by the unique electronic and spatial properties of nanocavities due to the ramified π-electron systems. This provides an opportunity for the formation of endohedral complexes containing non-covalently bonded atoms or molecules inside fullerenes and nanotubes. The guest species are exposed to the force field of the nanocavity, which can be described as a combination of electronic and steric requirements. Its action significantly changes conformational properties of even relatively simple molecules, including ethane and its analogs, as well as compounds with C−O, C−S, B−B, B−O, B−N, N−N, Al−Al, Si−Si and Ge−Ge bonds. Besides that, the cavity of the host molecule dramatically alters the stereochemical characteristics of cyclic and heterocyclic systems, affects the energy of pyramidal nitrogen inversion in amines, changes the relative stability of cis and trans isomers and, in the case of chiral nanotubes, strongly influences the properties of *R*- and *S*-enantiomers. The present review aims at primary compilation of such unusual stereochemical effects and initial evaluation of the nature of the force field inside nanotubes and fullerenes.

## 1. Introduction

The history of nanotubes [1,2] and fullerenes [3] spans several decades; the most important stages in understanding of their nature and properties are well represented in recent reviews and books [4,5,6,7,8,9,10]. The main directions of practical application of such nanoobjects are connected with creation of new materials [4,7,8,9,10,11,12,13,14], applications in pharmacy and medicine [15,16,17,18,19,20,21,22,23,24] and wastewater treatment [25]. These possibilities resulted mainly from the formation of endohedral complexes of fullerenes and nanotubes; the first example, synthesis of lanthanum complexes of C_60_, was described in [26]. Soon after, by using a suitable solvent, supercritical CO_2_, molecular surgery, and plasma ion irradiation methods, a great number of endohedral fullerenes and nanotubes, with metals, nitrogen, hydrogen, boron, noble gases, halogens, sulfur, radioactive isotopes, metal nitrides, water, ammonia, methane, methanol, aromatic hydrocarbons, cyclohexane, heterocycles, metallocenes and biomolecules in their cavities, were obtained [27,28,29,30,31,32,33,34,35,36,37,38,39,40,41,42,43,44,45,46,47,48,49,50,51,52,53,54,55,56,57,58,59,60,61,62,63,64]. The endohedral fullerene complexes inside nanotubes are another type of such species; a recent example is presented in [65]. The structure and properties of hybrid molecule–nanotube and fullerene systems were investigated with both experimental and theoretical methods. In addition to the practical use (such clusters, with unique magnetic and optical properties, are interesting as conductors and semiconductors, effective transporters of drugs in biological systems, and for hydrogen and methane storage) it is important to note that due to the extensive charge transfer between the carbon cage and the guest molecule, the latter acquires an electric charge [27,29,35,36,37,43,52,62,66]. Besides that, weak and van der Waals interactions play an important role in the molecular encapsulation [66]. It was shown that flat molecules (benzene, coronene, perylene) tend to align their molecular planes with the nanotube axis. In this case, the chirality of the tubes does not matter, but the tube has an influence on the intermolecular distances for confined species. It was also established that the encapsulation of hydrocarbons in nanotubes is energetically profitable [66]. In the case of armchair (n,n) single-walled carbon nanotubes (SWCNTs) and encapsulated hydrohalogens, the final orientation of the confined molecule depends on the internal diameter of the SWCNT’s hollow space and the halogen’s nature (F, Cl, Br, I) [67].

Guest species inside nanotubes can move under the action of different forces. This phenomenon is experimentally confirmed for “peapod” systems (fullerenes and endohedral fullerenes in SWCNTs) [27]. Besides that, unique conditions in nanotube cavity make possible numerous chemical reactions. For example, theoretical investigation predicted decreasing of the activation barrier of the Menshutkin S_N_2 reaction inside nanotubes [68]. On the other hand, the calculated energy barrier for Cl^−^ exchange in the S_N_2 reaction within nanotubes was higher by 6.6 kcal/mol in comparison with the gas phase [69]. Later, it was shown that SWCNTs may be used as effective nanoreactors for preparative syntheses of inorganic [70] and organic [71,72,73,74,75] products with high yields; it is also possible to achieve enantiomeric excess of the products by using a racemic mixture of *P* and *M* enantiomers of (n,m) chiral nanotubes [76].

All of the foregoing clearly shows that the domestic area of fullerenes and nanotubes can be viewed as the action of a solvent with specific properties. If that is true, then the question arises: how does that affect the conformational preference and other stereochemical properties of the encapsulated molecule?

As indicated by C. Reichardt, “The interactions between species in solvents (and in solutions) are at once too strong to be treated by the laws of the kinetic theory of gases, yet too weak to be treated by the laws of solid-state physics” [77]. This statement is one of the main components of solute–solvent interaction research strategy. Its further development has resulted in two solvent models: implicit (continuum) and explicit (discrete); their comparative effectiveness has been discussed in recent publications [78,79,80,81,82,83,84]. In this connection, the endohedral complexes of fullerenes and nanotubes can be viewed as a variety of explicit models. Various approximations based on DFT are usually used as an appropriate computational technique for simulation of the structural, electronic, and conformational properties of such systems. Some of the most useful among them are the PBE, hybrid PBE [85] and wB97XD methods [86]. The author’s investigations are connected with the approximation PBE/3ζ (PRIRODA program [87]) that correctly describes the structural, thermochemical, and polar characteristics of endohedral complexes of fullerenes [88,89,90,91]. The 3ζ triple-split basis set [92] is a full-electron non-relativistic Gauss-type set containing an accelerating aux-part and polarization functions. This allowed to get a comparative analysis of structural change of different compounds in nanocavity using a single proven method of calculation.

The first example of conformational investigation of simple molecules in the nanocavity was devoted to the examination of the torsion motion of H_2_O_2_ inside (5,5) and (6,6) armchair SWCNTs, using B3LYP/6-31G(d, p) approximation [93]. Two types of orientation of the O−O bond of the guest molecule inside nanotubes—along and perpendicular to the tube axis—were considered. The H−O−O−H torsion angle τ was varied from 0 to 180° with 10° steps. It was established that, in the case of (5,5) SWCNT, the orientation along the axis corresponds to the more stable endohedral complex compared to the perpendicular one. The main minimum on the potential energy surface (PES) in this case belongs to the conformer with τ = 160°, while for the free H_2_O_2_ molecule it corresponds to the form with τ = 120°. For the adduct H_2_O_2_@(6,6), with a larger diameter of the nanotube, both orientations of the guest molecule remain stable; the main minimum for the complex with an O−O bond oriented along nanotube axis corresponds to the conformer with τ = 70°. The effect of the weak O−H∙∙∙π interaction on the binding of the guest molecule is discussed. Polarization of the nanotube in such complexes was also shown.

In view of the foregoing, the objective of the present work is to discuss the conformational behavior and other stereochemical properties a range of compounds: inorganic species, ethane and its analogs, alcohols, ethers, compounds with double bonds, enantiomers, cyclic and heterocyclic molecules—inside nanotubes and fullerenes in comparison with free state or usual solutions. The main focus is on the influence of the size and chemical composition of nanoobjects as well as of SWCNTs’ chirality and diameter of fullerenes on the stability of preference conformation and chemical structure of the guest molecule.

## 2. Conformational Behavior of Ethane and Its Analogs in Nanotubes

Ethane is the classic object of conformational investigations. After Orville-Thomas’ fundamental book [94], there was a noticeable increase in the number of papers, which raised the question: why is the staggered form of ethane more stable compared to the eclipsed one? According to the concepts of theory, the nature of the rotational barrier about a simple bond is generally determined by exchange-repulsion, electrostatic, steric and hyperconjugation effects. Their relative contributions have been widely discussed in the literature [95,96,97,98,99,100,101,102,103,104,105,106,107,108,109,110]. On the other hand, spectral data indicate that ethane molecules adsorbed on the surfaces of silver, indium, and potassium exist in the eclipsed conformation [111]. Computer simulation shows that this is not the only example of such an unusual conformational behavior of alkanes. For example, nanoscale confinement (protein binding sites or carbon nanotubes) significantly changes conformations of butane, hexane and tetracosane from trans to gauche-rich helical form [112,113,114]. It is also known that linear alkanes are adsorbed on the surface of zeolites in a highly coiled conformation [115]. On the other hand, direct observation of conformational changes of a single molecule using transmission electron microscopy (TEM) [116] indicates the presence of only 6% of the CF_2_−CF_2_ and 20% of CH_2_−CH_2_ bonds in gauche conformation in a single perfluoroalkyl fullerene molecule inside SWCNT [117]. This qualitatively meets the known conformational data for alkyl chains of free hydrocarbon molecules [94]. However, it should be stressed that citing data refer to the alkane chain that contains fullerenes. Because of this additional factor, the flexibility of the chain links inside the nanotubes may be considerably restricted.

### 2.1. Ethane

The investigation of adsorption and diffusion properties of ethane within SWCNTs was presented in [118,119,120,121,122,123]. It has been established that nanotube geometry in some cases plays a definite role in the adsorption activity. The selectivity of adsorption of a binary mixture of ethane and ethylene was also studied. On the other hand, the rotational barriers of methyl-sized groups depend on van der Waals interactions and may vary under the influence of the nanosurface [124]. The first example of conformational transformation of ethane in the nanotube cavity was presented in [125].

Surprisingly, during the calculation (DFT approximation PBE/3ζ), the initial staggered form of ethane inside SWCNT (4,4) spontaneously converted to the eclipsed conformation. Its Hessian, in contrast to the staggered form, did not contain imaginary frequencies (Figure 1).

The relative stability of the staggered form of free C_2_H_6_ (ΔG^≠^_298_) is 2.5 kcal/mol. It should be noted that this value is slightly underestimated in comparison with data from the literature (2.8–3.04 kcal/mol [94]). In the case of encapsulated ethane, the eclipsed form is more stable than the staggered form at 0.4 kcal/mol. Moreover, both encapsulated forms have a negative electric charge (−0.53), the shortened C–C bond length (r_C–C_ 1.502–1.505 instead of 1.531–1.544 Å for the free molecule) and decreased Mulliken’s bond order (O_C–C_ 0.78–0.79 instead of 1.00–1.02). The guest molecule is oriented along the symmetry axis of the nanotube. The distance between hydrogen atoms of ethane and the SWCNT sceleton is 1.7 Å [125].

This relationship was confirmed for other endohedral clusters of ethane with different types of SWCNTs (Table 1, Figure 2) [126,127,128].

A general characteristic of all clusters is the decrease in C−C bond length with simultaneous reduction of its Mulliken bond order and appearance of positive or negative electric charge on the guest molecule in accordance with the conclusions of [27,29,35,36,37,43,52,62,66]. SWCNTs with zig-zag geometry, (6,0)-short and (6,0)-long, differ in length (7.11 and 9.34 Å respectively). All endocomplexes demonstrate an evident preference for the eclipsed (or nearly eclipsed) conformation of ethane, although the last cluster with host tube (6,0) with BN fragments (BN nanotubes have been studied in [129,130,131]) has shown a tendency to reduce the energy differences between both forms, in comparison with carbon-analog: nanotube (6,0)-long [127]. The example of complex C_2_H_6_@(6,0)-CBN also demonstrates that, with regard to the van der Waals spheres, the inner cavity of the nanotube is densely filled with the guest molecule (Figure 2).

### 2.2. Propane

Computer simulation of propane inside two model SWCNTs, (4,4) and (6,0) (PBE/3ζ), shows that staggered conformation in this case is not realized: during the optimization, the guest molecule converts into the fully eclipsed form that cannot exist outside the tube (Figure 3) [132]. This form is more stable than the partially eclipsed conformation (transition state, TS) by 1.6 and 1.2 kcal/mol for the endocomplexes with (4,4) and (6,0) SWCNTs respectively (ΔG^≠^_298_); meanwhile, the calculated barrier of internal rotation for the free propane is 3.0 kcal/mol in favor of the staggered conformer.

The guest species are characterized by shortened C−C bonds and a small negative electric charge (−0.37–−0.39); the inner cavity of SWCNT is completely filled with the guest molecule [132].

### 2.3. 2,2,3,3-Tetramethylbutane

According to the electron diffraction data, the structure of gaseous 2,2,3,3-tetramethylbutane (hexamethylethane, C_8_H_18_) corresponds to the staggered form. The central C–C bond length is 1.582 Å [133]; our calculation gives the value 1.590 Å. In this case, the relative stability of the staggered conformer is 8.6 kcal/mol (PBE/3ζ) [134]. However, in endocomplex C_8_H_18_@(5,5), the preferred conformation of the guest molecule is nearly eclipsed (Figure 4).

The relative stability of the eclipsed form is 0.9 kcal/mol (ΔG^≠^_298_); the length of the central C−C bond is 1.522 Å (shorter by 0.068 Å than in the free alkane). The encapsulated molecule acquires a negative electric charge (−0.93) [134].

### 2.4. 2,2-Dimethylpropane

2,2-Dimethylpropane (neopentane) is a convenient model system for investigation of the interconnection between the rotational barrier and the molecular environment around the rotator [124,135]. The experimental barrier to internal rotation (ΔG_298_^≠^) is 4.3 kcal/mol [136,137]. However, the conformational behavior of neopentane inside nanotubes differs significantly from that of ethane and its analogs discussed above. Its main feature is a substantial increase in the internal rotation barrier in the case of endocomplexes with SWCNTs of small diameter; TS corresponds to the eclipsed form. According to the PBE/3ζ approximation, the value of ΔG_298_^≠^ for cluster C_5_H_12_@(5,5) with diameter 6.8, Å equals 11.1 kcal/mol, in comparison with 3.8 kcal/mol for the free molecule; the corresponding value for the barrel-shaped SWCNT (7,0) with diameter 9.6 Å is 4.6 kcal/mol. In both cases, the guest molecule has a negative charge (−0.72 and −0.49 for the ground state) [138].

### 2.5. Fluoroethanes

The conformational behavior of fluoroethanes in the SWCNT cavity is not different from that observed for ethane itself: in the case of a small nanotube diameter, the preferred conformation corresponds to the eclipsed form. Its relative stability varies from 0.8 to 2.3 kcal/mol (Figure 5 and Table 2). An increase in nanotube diameter leads to the preference for a staggered form that, however, remains less stable than for the free molecule [139,140,141,142,143].

In all cases, the guest molecule has a negative electric charge; besides that, cluster C_2_H_5_F@(4,4) is characterized by a reduced value of Mulliken C–C bond order (O_C−C_).

### 2.6. Ammonia Borane

Ammonia borane (borazane) H_3_B←NH_3_ has an ethane-like structure; according to microwave measurements, the barrier to internal rotation about the B←N bond equals 2.008–2.047 kcal/mol [144]. The main reasons for the relative stability of staggered form have been discussed in [145]. On the other hand, ammonia borane is a valuable source of hydrogen [146,147]. The creation of effective hydrogen storage is connected with nano-encapsulated H_3_B←NH_3_ [148,149]. In particular, such systems include ammonia borane clusters with polypyrrol nanotubes [150]. In this connection, the investigation of borazane endocomplexes is an important direction in nanochemistry.

The conformational properties of ammonia borane in the SWCNT cavity are very close to those of ethane. Clusters H_3_B←NH_3_@(4,4), H_3_B←NH_3_@(7,0) and H_3_B←NH_3_@(8,0) demonstrate the preference for the eclipsed form; its relative stability reaches 0.4–0.8 kcal/mol (ΔG^≠^_298_). In some cases, both forms are degenerate in energy [151].

A comparison of free and encapsulated molecules of H_3_B←NH_3_ (Figure 6) demonstrates that boron atomic charge in the cluster has changed its sign and Mulliken B←N bond order increased almost twofold; all of this indicates the significant redistribution of electron density in the guest molecule.

### 2.7. Disilane and Digermane

Different computational approaches confirmed hindered rotation in disilane, its staggered form being more stable than the eclipsed one [100,152,153,154]. The calculated rotational barrier (PBE/3ζ, ΔG^≠^_298_, 1.3 kcal/mol [155]) is very close to the one experimentally measured (1.26 kcal/mol [156]). At the same time, computer simulation demonstrates that the height of the barrier in complex Si_2_H_6_@(8,0) is only 0.4 kcal/mol in favor of the staggered form. In contrast to ethane clusters, the Si–Si bond length of encapsulated disilane is increased by 0.071–0.045 Å in comparison with the free molecule. The growth of the nanotube diameter in the case of cluster Si_2_H_6_@(6,6) increases the value of ΔG^≠^_298_ to 1.7 kcal/mol (PBE/3ζ, [155]).

The experimental barrier of rotation about the Ge-Ge bond in digermane corresponds to 1.2 kcal/mol [157]. On the other hand, this compound is widely used in epitaxial technology for the creation of mono- or polycrystalline thin films on the surfaces of silicon [158,159,160,161], tin [162], and germanium [163,164], with remarkable semiconducting properties. In this connection, endohedral nanocomplexes containing Ge_2_H_6_ are interesting as an important component in this process.

The conformational behavior of digermane in cluster Ge_2_H_6_@(8,0) demonstrates a preference for the eclipsed form (ΔG^≠^_298_ 0.8 kcal/mol, in comparison with 1.1 kcal/mol in favor of the staggered form of the free molecule, PBE/3ζ) [165]. As in the previous case, the encapsulated molecule is characterized by a 2% growth in the length of Ge-Ge bond and a strong negative electric charge (−1.37). All other examined clusters—Ge_2_H_6_@(5,5), Ge_2_H_6_@(6,6) and Ge_2_H_6_@(7,7)—with a greater nanotube diameter exhibit a preference for the staggered conformation (ΔG^≠^_298_ 1.1–1.5 kcal/mol) and electric charge (−0.12–−0.97) on the guest molecule [165].

Thus, the action of the force field inside the nanocavity, understood as a combination of electronic and steric requirements, shifts the conformational equilibria of ethane and its analogs inside nanotubes of relatively small diameter (with the exception of neopentane) to the eclipsed form. This is accompanied by a significant change in the central bond length and the bond order.

## 3. Conformational Behavior of Other Acyclic Molecules in Nanotubes

### 3.1. Hydroxyborane

In terms of structural chemistry, acyclic derivatives of boron and boronic acids are interesting because of the partially double nature of the B-O bond; as a result, their preferred conformation corresponds to the planar form [166,167]. The rotation barrier lies within 8.5–13.8 kcal/mol [168,169,170]. According to the PBE/3ζ approximation, the orthogonal conformation—TS for the molecule of hydroxyborane, H_2_B-OH—is less stable than the planar form (ΔG^≠^_298_) by 13.9 kcal/mol (Figure 7) [171].

However, in the case of cluster H_2_B-OH@(6,0), the value of ΔG^≠^_298_ becomes only 3.8 kcal/mol (3.66 times lower); the increase in the SWCNT diameter, as in the case of H_2_B-OH@(6,6), leads to a higher barrier of rotation close to the free molecule (13.4 kcal/mol). The guest molecule in both cases is characterized by electric charge and—for cluster H_2_B-OH@(6,0)—by a 1.4% decrease in the length of the B-O bond [171].

### 3.2. Diborane (4)

A hypothetical diborane (4) is the boron analog of ethylene. Due to the nature of the boron–boron bond, it represents a model system with non-classical multi-centered bonds [172,173]. The PES of this compound contains three stationary points that correspond to the structures **A** (C2v symmetry), **B** (D2d), and **C** (D2h) [174] (Figure 8).

The first matches the global minimum; the orthogonal conformer **B** is less stable than **A** by 2.9 kcal/mol [CCSD(T)/6-311++G (d,p)] [174] or by 3.0 kcal/mol (PBE/3ζ [175]). It turns into its inverted form (**B***) via the TS—a planar structure **C** with a barrier of 17.9 kcal/mol [175].

All examined SWCNT clusters demonstrate a significant shortening of the B–B bond length in all forms and a relatively high negative electric charge on the guest molecule. Besides that, form **A** belongs to the main minimum on the PES, but differences in energy between **A** and **B** increase 2–4.5 times. The height of the rotation barrier about the B-B bond in endocomplexes is lowered compared with a free molecule; for clusters B_2_H_4_@(4,4) and B_2_H_4_@(7,0), it lies in the interval 3.1–5.7 kcal/mol. The growth of the nanotube diameter in complexes B_2_H_4_@(5,5) and B_2_H_4_@(8,0) leads to an increase in this value (14.1–17.4 kcal/mol relative to the form **B**) [175].

### 3.3. Dialane (4)

Dialane (4), Al_2_H_4_, in contrast to diborane, really exists in two forms: salt-like structure **A** and orthogonal **B** [176,177]. The first corresponds to the global minimum on the PES. The last may be converted into invertomer **B*** via TS—planar conformation **C** (Figure 9) [178].

According to the results of computer simulation, the conformational equilibrium between **B** and **C** forms in clusters Al_2_H_4_@(5,5), Al_2_H_4_@(8,0) and Al_2_H_4_@(9,0), contrary to the free molecule, is completely shifted to the planar form **C**: the appropriate ΔG_298_^≠^ values lie in the interval 2.3–4.5 kcal/mol; ΔE_0_^≠^ corresponds to 1.4–4.2 kcal/mol (Figure 10). In all cases, the guest molecule has a strong negative electric charge [178].

The authors [107] suggested a decisive role of all seven molecular orbitals in the origin of the rotational barrier in ethane. But the common number of MO’s in nanoclusters is very high. That is why we analyzed only HOMO and LUMO orbitals of dialane (4), as well as SWCNT (8,0) and cluster Al_2_H_4_@(8,0), using PBE/3ζ and PBE/cc-pVDZ approximations [178] (Figure 11).

The main difference between the free molecule and cluster is connected with a notable increase in the HOMO energy in the last case; this is also true for the nanotube itself. Thank to this, the energy gap ΔE for complex Al_2_H_4_@(8,0) is decreased by 28–34 times compared to free dialane. According to the molecular diagrams, HOMO is localized on the carbon atoms of nanotubes; the highest-energy MO connected with the guest molecule (the planar form of dialane) corresponds to the HOMO-3 level with the energy of −5.3350 eV (PBE/3ζ). Similar results were obtained using PBE/cc-pVDZ approximation.

All this points to the significant transformation of the orbital structure of the endocomplex, which may be a possible reason for the change in conformational behavior of the guest molecule.

### 3.4. Hydrazine

Hydrazine, which is widely used in organic synthesis and in medicine, demonstrates the influence of unshared electron pairs on the rotation barrier about N–N bond. Its conformational equilibrium includes gauche-form **A** (global minimum), staggered form **B** (local TS) and eclipsed form **C** (main TS); the torsion angle τ in conformer **A** equals 91.5° [179,180,181,182]. The energy differences between conformations **A**-**B** and **A**-**C** (ΔG_298_^≠^) are 1.8 kcal/mol and 7.9 kcal/mol respectively (PBE/3ζ, Scheme 1) [183].

The force field of SWCNTs substantively changes the conformational behavior of guest molecule (PBE/3ζ) [183]. Clusters N_2_H_4_@(4,4) demonstrate a significant decline (to 4.6 kcal/mol) of ΔG_298_^≠^ (forms **A**-**C**). In the case of SWCNTs (4,4) and (6,0), the main minimum of hydrazine belongs to the conformer **B** with ΔG_298_^≠^
**B**-**C** 1.1–2.2 kcal/mol. The increase in the diameter of the nanotube, as for endocomplex N_2_H_4_@(6,6), results in an equilibrium close to the free hydrazine. In all cases, the guest molecule has a positive or negative electric charge [183].

### 3.5. Methanol and Methanethiol

Methanol and methanethiol are characterized by a hindered rotation with the potential barrier between gauche (**A**, minimum) and eclipsed (**B**, TS) forms at 1.07 and 1.27 kcal/mol respectively (Figure 12) [94,184]; according to PBE/3ζ approximation, the appropriate ΔG_298_^≠^ values are 1.0 and 1.2 kcal/mol [128,185]. However, in the cavity of SWCNT (6,0), their conformational behavior has significantly changed.

In the case of cluster CH_3_OH@(6,0), minimum on the PES belongs not to the form **A**, which becomes the main TS, but to the conformer **C,** with a torsion angle of HOCH 26.4° (PBE/3ζ, Figure 12). The main (**A**) and a local (**B**) TS are less stable by 0.27 and 0.03 kcal/mol respectively (ΔG_298_^≠^) [185]. Endocluster CH_3_SH@(6,0) is characterized by the torsion angle HSCH 22.2° and ΔG_298_^≠^ values for **A** and **B** forms of 2.1 and 0.6 kcal/mol. In both cases, the guest molecule has a positive electric charge (0.3–0.7). Again, as in the previous examples, all this indicates the important role of *n*-electron pairs in the guest molecule, affecting changes in its conformational preference outside and inside nanotubes.

### 3.6. Dimethyl Ether

The molecule (CH_3_)_2_O belongs to the systems with two internal rotors; the experimental barrier of rotation between **A** and **B** forms is 2.60 kcal/mol (near 900 cm^−1^) (Figure 13) [186,187,188,189]. The origin and nature of the barrier, in particular the role of lone pairs, is widely discussed in the literature [190,191,192,193]. However, in the SWCNT cavity, the conformational behavior of this compound is significantly changed. In clusters (CH_3_)_2_O@(4,4), form **A** is not realized: during the optimization it converts to the unusual conformer **C**, which corresponds to the minimum and cannot exist outside the tube, turning back into form **A** (Figure 13) [194]. According to the PBE/3ζ, approximation the valence angle COC in the free molecule of dimethyl ether (form **A**) is 111.2°, which is very close to the experimental value (111.5° [188]). However, for form **C** in cluster (CH_3_)_2_O@(4,4), it rises to 127–130° [194]. Meanwhile, form **B** in this case also corresponds to the TS; the value of ΔG_298_^≠^
**B**-**C** (1.3–1.9 kcal/mol) is lower than that for the transition **A**-**B** of the free molecule (2.5 kcal/mol).

In the case of endocomplex (CH_3_)_2_O@(6,0), the angle COC in conformer **C** rises to 180° and the molecule takes the form of a rod with an eclipsed conformation of methyl groups (Figure 13). The appropriate transition state corresponds to the rod-shaped form, with a staggered conformation of methyl groups and ΔG_298_^≠^ 3.5 kcal/mol.

In all cases, the guest molecule has positive or negative charge and shortened C-O bonds [194].

## 4. Conformational Properties of Simple Molecules in Fullerenes

### 4.1. Ethane and Its Analogs

The conformational behavior of ethane inside fullerenes of small diameter was studied on the example of cluster C_2_H_6_@Si_20_ [195]. PES, in this case, contains tree stationary points: partially eclipsed **A** (global minimum), eclipsed **B** (main TS) and staggered **C** (local TS) (Figure 14).

Differences in energy between these forms (ΔG_298_^≠^, PBE/3ζ) account for 1.2 (**A**-**B**) and 0.4 (**A**-**C**) kcal/mol. Hence, compared to the free ethane, the internal rotation barrier has become two times lower. The guest molecule possesses a strong positive electric charge (1.3–1.4) and is characterized by shortened C−C bond length and a decline in its Mulliken bond order [195].

Endoclusters of ethane-like molecules with fullerenes C_60_, C_70_, C_80_ and their heteroanalogs, in contrast to the previous example, are characterized by only two stationary points on the PES: staggered and eclipsed forms. The first corresponds to the minimum and the second to the TS [196,197,198,199,200,201,202]. Ethane inside the C_60_ cavity exhibits a higher barrier of internal rotation and shorter C−C bond length in comparison with the free molecule [196]. The same is true of its analogs; in some cases, the height of the barrier grows by more than threefold (Table 3). This increase in diameter of fullerenes (C_60_, C_70_, C_80_ and Si_60_) naturally leads to the decrease in ΔG_298_^≠^. In the case of B-N fullerenes (their peculiarities have been discussed in [131]) this dependence is more mixed. In particular, in a series of endocomplexes, C_2_F_6_@C_60_, C_2_F_6_@C_12_B_24_N_24_ and C_2_F_6_@B_36_N_24_, the value of ΔG_298_^≠^ initially decreases and then increases, while it is still lower than that of the C_60_ cluster [198]. This influence of chemical composition is also valid for the H_3_B←NH_3_ endocomplexes with fullerenes containing both 60 and 80 atoms in their shells [201]. Meanwhile, in series of C_2_F_6_ clusters with 80-atomic fullerenes, the barrier for the (BNC)_80_ system is, vice versa, higher than those of C_80_ and (BN)_80_ complexes (Table 3). All guest molecules have electric charge; in the case of C_5_H_12_ clusters, it is rather large and changes the sign for different fullerenes [200]. Mulliken bond order for the central bond (O), with the exception of several H_3_B←NH_3_ endocomplexes, depends little on the size and chemical composition of the fullerenes.

### 4.2. Methanethiol

The conformational equilibrium of methanethiol in endocomplexes with fullerenes, as for the free molecule, is characterized by gauche (minimum) and eclipsed (TS) forms (see part 3.5). But the height of the rotational barrier about the C-S bond (ΔG_298_^≠^), in this case, is increased: 2.0 and 2.7 kcal/mol for the clusters CH_3_SH@C_60_ and CH_3_SH@C_80_, respectively, in comparison with 1.2 kcal/mol for the free molecule (PBE/3ζ) [203]. As in previous cases, the guest molecule has a certain electric charge (−0.5).

## 5. Conformational Behavior of Saturated Cyclic Molecules inside Nanotubes and Fullerenes

There are very few examples devoted to the behavior of saturated cyclic molecules, in particular, cyclohexane [33] and octasiloxane [27], inside SWCNTs. These are largely focused on the orientation of the guest molecule as a whole in the nanocavity. It should be noted, however, that in view of the structural transformations of acyclic molecules that were discussed in previous sections, the conformational behavior of cyclic systems inside nanoobjects also has a number of distinguishing features.

### 5.1. Cyclohexane in Nanotubes

It is well known that, at usual temperatures, molecules of cyclohexane exist in conformational equilibrium between chair (C, main minimum) and twist (Tw, local minimum) forms; the TS corresponds to the half-chair conformation (Scheme 2) [94].

The conformational behavior of cyclohexane in the SWCNT cluster C_6_H_12_@(8,0) is characterized by the absence of a Tw-form: the C-conformer directly converts into its invertomer C* over TS that corresponds to the conformation close to the flattened semi-planar form; the height of the barrier (ΔG_298_^≠^, PBE/3ζ, 10.4 kcal/mol) is very similar to that for the free cyclohexane (10.5 kcal/mol). The guest molecule has a positive charge (0.62) and is oriented in the direction perpendicular to the axis of the nanotube [204] (Figure 15).

The conformational behavior of cyclohexane in cluster C_6_H_12_@(5,5) is the same, except that TS in this case corresponds to the Tw-form, the height of the barrier is 6.8 kcal/mol and the guest molecule has a negative charge (−0.50) [204].

### 5.2. 1,3-Dioxane in Nanotubes

It is known that 1,3-dioxanes—very attractive heteroanalogs of cyclohexane—belong to the classical objects of conformational analysis [94]. Besides that, due to their wide range of pharmacological action, they may be used for the creation of new drugs [205,206] and also as reagents in fine organic synthesis [207,208,209,210,211]. All this makes it relevant to consider the conformational properties of these compounds inside nanotubes, from the perspectives of nanoreactors [68,69,70,71,72,73,74,75] and drug delivery systems [15,19].

The PES of unsubstituted 1,3-dioxane, C_4_H_8_O_2_, contains tree minima (C—the main, 2,5-Tw and 1,4-Tw-forms—the local) (Scheme 3) [94,212].

However, in a SWCNT cavity of small diameter (≤8.4 Å), the relative population of these forms significantly changes. In the case of cluster C_4_H_8_O_2_@(6,6), the 1,4-Tw conformer is not realized, and PES contains only C (main minimum) and 2,5-Tw-forms with ΔG_298_^0^ 3.1 kcal/mol, which is significantly lower than for free 1,3-dioxane (5.0 kcal/mol). In the case of clusters C_4_H_8_O_2_@(5,5) and C_4_H_8_O_2_@(8,0), the main minima correspond to the 2,5-Tw-conformer, and are 3.0 and 0.3 kcal/mol, respectively, more stable than the C-form; the last remains only as a local minimum. At the same time, the barrier of interconversion (ΔG_298_^≠^) is reduced by up to 5.1 kcal/mol in comparison with 9.7 kcal/mol for free 1,3-dioxane (PBE/3ζ) [213]. Cluster C_4_H_8_O_2_@(7,0) demonstrates the presence of only one form—2,5-Tw; the C-conformer in this case is not realized (Figure 16). The van der Waals spheres show that the inner cavity of the nanotube is densely filled with 1,3-dioxane. In all cases, the guest molecule has a negative electric charge.

It should be stressed that the conformational behavior of substituted 1,3-dioxanes in usual solvents is not accompanied by the inversion of stability between the chair and any other form; in all cases, C-conformer remains the main minimum, and the role of the medium is relegated only to the shift of the conformational equilibrium towards the chair-form with more or less polarity, due to the axial or equatorial orientation of polar substituents or with lower steric restriction [214,215]. In sharp contrast, the conformational properties of unsubstituted 1,3-dioxane in the SWCNT cavity are caused by the action of a new driving force, which can dramatically alter conformational properties of the guest cyclic molecules in comparison with those in the gas phase or in the usual solvents.

### 5.3. 1,3-Dioxa-2-Silacyclohexane in Nanotubes

The stereochemistry of silicon-containing saturated heterocycles is an integral part of organosilicon chemistry; the introduction of silicon atoms changes the geometry and electronic features of the cyclic system [216]. According to the results of gas-phase electron diffraction, the molecular structure of 2,2-dimethyl-1,3-dioxa-2-silacyclohexane corresponds to the flattened chair [217]. Theoretical investigation using Hartree–Fock (HF) and DFT approximations led to the conclusion of a rather low barrier of interconversion between two C-forms, including a local minimum of 2,5-Tw, which is very close in energy to the TS (Scheme 4) [212,217,218,219]. In the case of free 1,3-dioxa-2-silacyclohexane, C_3_H_8_SiO_2_ (R=H), the corresponding ΔG_298_^≠^ is 4.1 kcal/mol (PBE/3ζ) [220].

It was found that the main minimum of the guest molecule in cluster C_3_H_8_SiO_2_@(5,5) belongs to the structure close to the half-chair (CH) conformer, not to the C-form. But other than that, the details of conformational behavior of encapsulated molecule remain the same; the barrier of interconversion is slightly higher than for the free molecule (5.1 kcal/mol, Scheme 5). Form 2,5-Tw remains the local minimum on the PES [220].

In the case of cluster C_3_H_8_SiO_2_@(8,0) the main minimum corresponds to the C-conformer. Equilibrium C ↔ 2,5-Tw in this case is described by the relatively low energy differences between these forms (2.6 kcal/mol in comparison with 3.3 kcal/mol for the free silicon ester and previous cluster). The barrier of interconversion is slightly increased (4.8 kcal/mol). In all cases the guest molecule has a negative electric charge (−0.25–−0.87) [220].

Thus, SWCNTs cavity may change the conformation corresponding to the minimum as well as alters certain energy characteristics of conformational equilibrium of six membered cyclic silicon esters.

### 5.4. Hexahydropyrimidin-2-One in Nanotubes

Substituted hexahydropirimidine-2-ones belong to the promising class of cyclic urea derivatives with a well-defined biological activity [221] and as reagents in organic chemistry [222,223]. According to the X-ray investigation their structure corresponds to the semi-planar form, or C-conformer with a strong flattened heteroatomic part [224]. Conformational transformation of the free molecule of hexahydropyrimidin-2-one, C_4_H_8_N_2_O, established by HF/6-31G(d) and PBE/3ζ methods includes two degenerated in energy flattened C-conformers, 2,5-Tw as a local minimum and TS that corresponds to the conformation very close in energy to the 2,5-Tw form (Scheme 6) [225,226].

Conformational equilibrium of hexahydropirimidine-2-one in cluster C_4_H_8_N_2_O@(5,5) exhibits the absence of 2,5-Tw form as a local minimum: it becomes a TS. Besides this conformation PES contains two degenerated in energy flattened C-forms with the increased barrier of interconversion (5.4 kcal/mol in comparison with 3.3 kcal/mol for the free cyclic urea, PBE/3ζ). The guest molecule acquires a negative electric charge (−0.66). In the case of cluster C_4_H_8_N_2_O@(8,0) PES besides C-form and TS includes 2,5-Tw conformer as a local minimum; the value ΔG_298_^0^ between them (3.1 kcal/mol) is higher than for the free molecule (2.4 kcal/mol), but the barrier of interconversion (3.5 kcal/mol) changes very little. The encapsulated molecule has a positive charge (0.32) [226].

Thus, the influence of nanotubes on the conformational properties of unsubstituted cyclic urea is down mainly to the qualitative change in the nature of intermediate forms and to the variation in energy parameters of conformational equilibrium between flattened C-conformers.

### 5.5. 1,3,2-Dioxaborinane in Fullerenes

It was established recently that preferable conformation of 1,4-dioxane in exo adducts with C_60_ and C_70_ fullerenes is not a chair but a boat form. Such unusual result is connected with a rigidity of fullerene skeleton [227]. In this connection it would be interesting to assess the conformational properties of the guest cyclic molecule inside fullerenes. The first example of its kind was the simplest six-membered cyclic boronic ester: 1,3,2-dioxaborinane. It belongs to the valuable class of organoboron compounds which are widely used in organic syntheses [228,229,230,231] and are convenient models to assess the effect of heteroatoms in changing of the conformational properties of cyclohexane’s heteroanalogs [212]. The PES of 1,3,2-dioxaborinane itself (C_3_H_7_BO_2_) contains degenerated in energy conformers of sofa (Sf) and TS—2,5-Tw form (Scheme 7) [212].

It should be noted that stabilization of Sf-form in the free molecule is due to the p-π conjugation in the heteroatomic part of the cyclic ester because of the partially double nature of B-O bond [212]. However, endocomplex C_3_H_7_BO_2_@C_80_ demonstrates the conformational equilibrium between two unusual for this class of compounds boat (B) forms with ΔG_298_^≠^ 10.1 kcal/mol in comparison with 7.2 kcal/mol for the free molecule of cyclic boronic esters (PBE/3ζ, Scheme 8); the guest molecule has a negative charge (−0.74) [232].

The decrease in fullerene diameter leads to the more significant changes in conformational properties of the guest molecule. In cluster C_3_H_7_BO_2_@C_60_ the main minimum on the PES of cyclic ester belongs to the 1,4-Tw form that may convert to the C-conformer (local minimum) over TS—envelop (E) form (Scheme 9).

The value ΔG_298_^0^ between 1,4-Tw and C-conformers is 8.1 kcal/mol and the height of the barrier is 27.9 kcal/mol (3.9 times higher than calculated for the free ester); the guest molecule acquires a large positive charge (2.53 for the 1,4-Tw form) [232].

Thus, a combination of electronic and steric requirements that may be called as a force field inside fullerenes significantly changes conformational properties of even relatively simple cyclic molecule: forms like B, 1,4-Tw, E and C are never realized for 1,3,2-dioxaborinanes with sp^2^ boron atom in any solvent [212]. The sharp increase in population of twist-form is connected primarily with a considerable decline in linear size of the molecule inside fullerenes under the action of force field.

## 6. Nitrogen Pyramidal Inversion inside Nanotubes and Fullerenes

Recent years exhibit a significant interest to the investigation of pyramidal inversion in XH_3_ compounds [233]. One of the main challenges facing the theoretical spectroscopy is connected with a selection of ab initio and DFT methods that can ensure the accurate estimate the energy of nitrogen inversion barrier [234]. Authors of [235] suggested a benchmark set that comprises 24 high-level wave-function inversion barriers in different compounds; it is submitted that at least medium-sized triple-split-valence basis sets with at least one set of polarization functions should be used for the theoretical study of pyramidal inversion. Formally, method PBE/3ζ [87,92] quite meets those criteria. It should be stressed however that it is also underestimates the nitrogen inversion barrier in amines.

How an ordinary solvent affects the barrier of nitrogen pyramidal inversion? It was established that in polar medium this value depends on the number of solvent’s molecules in the close surroundings of the dissolved species. In particular, computer simulation of inversion in 3-methyltetrahydro-1,3-oxazine surrounded by the molecules of difluorodichloromethane using PBE/3ζ approximation (explicit model) revealed that the best match with experimental barrier was obtained for the case of four molecules in solvation shell [236].

### 6.1. Ammonia and Trimethylamine in Nanotubes

Experimental values of nitrogen pyramidal inversion barriers in ammonia and trimethylamine (ΔG_298_^≠^) are 5.8 and 7.5 kcal/mol respectively [237]. The appropriate results of PBE/3ζ has been understated (4.3 and 6.0 kcal/mol [238]). However, considering numerous challenges relating to the application of more complex methods and basis sets to the calculation of nanoobjects, it was possible to use the PBE/3ζ approach for the estimation of the inversion barrier of amines in SWCNTs. In the case of cluster NH_3_@(4,4) the value of ΔG_298_^≠^ was 10,9 kcal/mol (more than 2.5 times higher than for the free molecule). In contrast, the appropriate value for trimethylamine in cluster N(CH_3_)_3_@(6,6) was 4.8 kcal/mol (1.25 times lower than for the free molecule) [238]. It should be noted that the distance between guest species and carbon skeleton of SWCNT in both cases was the same (~2.5 Å). Thus, the value of inversion barrier in nanotube’s cavity as for the free molecule depends to some extent on the nature of amine. In both cases the guest molecule acquires a relatively small charge (up to −0.50).

### 6.2. Piperidine inside Nanotube

It is necessary to separate two dynamic processes relatively to piperidine (C_5_H_11_N): nitrogen pyramidal inversion and interconversion of the cycle. The first leads to the change in orientation of N-H proton (axial-equatorial, a–e) and the second over several steps—to the alternate conformation (Scheme 10) [239].

According to the NMR spectroscopy data the barrier of nitrogen pyramidal inversion (ΔG_298_^≠^) is 6.1 kcal/mol [240] and ΔG_298_^≠^ for the second process—10.4 kcal/mol [241]. So, the nitrogen inversion in this case requires lower energy compared to the conformational equilibrium of the cycle. Besides that, conformer C-e is more stable than C-a in the gas phase (0.72 kcal/mol) [242] and in nonpolar medium (0.2–0.6 kcal/mol) [243].

In cluster C_5_H_11_N@(6,6) the barrier of nitrogen inversion rises to 6.8 kcal/mol in comparison with 4.7 kcal/mol for the free molecule (PBE/3ζ) [244]. At the same time form C-a becomes more stable inside SWCNT; differences with C-e conformer (ΔG_298_^0^) is 2.6 kcal/mol compared to 0.6 kcal/mol in favor of form C-e for the free molecule. The encapsulated piperidine acquires a relatively large negative charge (−0.89 for the C-a form) [244].

### 6.3. Perhydro-1,3,2-Dioxazine inside Nanotubes

Perhydro-1,3,2-dioxazine (C_3_H_7_NO_2_) is the heteroanalog of cyclohexane with O–N–O ring fragment. It was firstly obtained 35 years ago [245]. Further study showed that this compound is conformationally rigid due to the relatively high barrier of nitrogen inversion in *N*,*N*-dialkoxyamines (21.7–24.6 kcal/mol) [246,247]. Detailed conformational analysis using MP2-RI/λ2 approximation revealed that the ΔG_298_^≠^ value for the direct transformation C-a ↔ C-e forms of perhydro-1,3,2-dioxazine is 23.3 kcal/mol (Scheme 11). Wherein conformer C-e is more stable at 2.5 kcal/mol (ΔG_298_^0^) [248].

In contrast to the free molecule cluster C_3_H_7_NO_2_@(6,6) demonstrates the preference of C-a form (ΔG_298_^0^ 2.1 kcal/mol, PBE/3ζ). The barrier of the nitrogen inversion is almost unchanged, compared to the free molecule (ΔG_298_^≠^ 23.4 and 21.7 kcal/mol respectively). The C-a form in the SWCNT cavity is slightly distorted [249]. The preference of C-a form is also observed in endocomplex C_3_H_7_NO_2_@(5,5); the corresponding ΔG_298_^0^ is 1.1 kcal/mol and ΔG_298_^≠^ for nitrogen inversion is rather close to the free perhydro-1,3,2-dioxazine (20.2 kcal/mol). In both cases the guest molecule acquires a positive charge (0.60–0.65) [249].

### 6.4. Ammonia and Trimethylamine in Fullerenes

Cluster NH_3_@C_60_ demonstrates almost the same barrier on nitrogen inversion as in the free molecule (4.4 kcal/mol); the N-H bond lengths remain practically unchanged and the guest molecule has a small negative charge (−0.30) [250]. Thus, the cavity of C_60_ is large enough to cause changes in the energy of inversion. In contrast, the inner space of fullerene in the case of N(CH_3_)_3_@C_60_ becomes rather small: the height of the barrier in this case is 3.4 times higher than for the free molecule (25.7 kcal/mol); the C-N bond lengths became shorter by 0.08 Å and the guest molecule acquires a positive charge (0.65 for the ground state and 1.14 for the TS). The increase of fullerene’s diameter leads to the decrease of ΔG_298_^≠^ for nitrogen inversion (10.3 kcal/mol for the cluster N(CH_3_)_3_@C_80_); the molecule of amine in this case has a negative charge (−0.83 for the ground state and −0.36 for the TS) [250].

Thus, the pyramidal inversion barrier for acyclic amines inside nanotubes and fullerenes depends on the nature of amine and on the size of nanocavity. In the case of cyclic amines in SWCNTs the inversion of the relative stability of axial and equatorial conformers is observed and, in the case of pyrimidine, the barrier of the nitrogen pyramidal inversion is increased in 1.4 times.

## 7. Recognition of the R- and S-Isomers by Chiral Nanotubes

It is well known that helicity is closely linked to chirality, because left-handed and right-handed helixes are non-superimposable to their mirror images. A distinctive feature of chiral nanotubes is the presence of a helical axis. This class of SWCNTs can be described by (n,m) chirality indexes (n ≠ m; m ≠ 0). A different chirality suggests that such objects create the individual enantiomeric pairs, also denoted as P and M nanotubes with a single set of values (n,m) [251]. Depending on the ratio between chirality indexes, they display features of conductors or semiconductors [252] and have valuable optical properties [253]. It was shown that the doping of chiral nanotubes with nitrogen diminishes the stress caused by the small diameter and the corresponding energy strongly depends on the tube helicity [254]. However, the most prominent quality of chiral nanotubes is their ability to form complexes with chiral molecules possessing different degrees of stability. In principle, this enable to provide asymmetric synthesis [76] and make possible a separation of left-handed from right-handed isomers by using, for example, chiral SWCNTs as stationary phases in chromatography. It is also possible to induce chirality in the chains of guest molecules [251,255,256]. In this connection, the relative stability of some R- and S-isomers in the cavity of chiral nanotubes was investigated using the PBE/3ζ approximation.

### 7.1. R- and S-Isomers of 1-Fluoroethanol Inside SWCNTs (4,4) and (4,1)

According to the previous analysis, both enantiomers of free 1-fluoroethanol (C_2_H_5_FO) participate in conformational equilibrium between **a** and **b** forms with differences in stability (ΔG_298_^0^) 0.3 kcal/mol in favor of **a** (Figure 17) [257]. This result is confirmed by [258].

It should be noted that both nanotubes—(4,4) and (4,1)—that were used as nanoobjects in clusters with 1-fluoroethanol have the same gross formula (C_80_H_16_), but SWCNT (4,4) is more stable than (4,1) at 0.6 kcal/mol (ΔG_298_^0^). Hence, these nanotubes can be viewed as structural isomers.

Molecules of 1-fluoroethanol inside SWCNTs (4,4) and M(4,1) convert into preferable conformation close to the eclipsed (Figure 18).

The energies of clusters R-C_2_H_5_FO@(4,4) and S-C_2_H_5_FO@(4,4) were predictably the same. However, endocomplex R-C_2_H_5_FO@M(4,1) was proven to be more stable than S-C_2_H_5_FO@M(4,1) at 4.4 kcal/mol, and in comparison with cluster C_2_H_5_FO@(4,4) at 12.8 kcal/mol (ΔG_298_^0^). This means that, despite the greater stability of SWCNT (4,4), cluster R-C_2_H_5_FO@M(4,1) exhibits a large energy gain which can serve as a theoretical basis to separate both enantiomers. Besides that, the guest molecule in all cases has a slight negative charge (up to −0.12) [257].

### 7.2. R- and S-Isomers of α-Alanine inside SWCNTs (n,m)

Conformational analysis of free α-alanine (C_3_H_7_NO_2_), carried out for S-isomer by rotating around the bond C(NH_2_)−C(O), indicates the presence of several stationary points on the PES; the main minimum corresponds to the form **A**, and the main TS to the conformation **B** (Scheme 12, Table 4) [259]. These results are quite consistent with the data of [260].

SWCNTs (5,5), (5,1) and (5,2), which were used as nanoobjects for the investigation of α-alanine properties in endocomplexes, differ in energy: as in the previous case (Section 7.1), the most stable is nanotube (5,5) (Table 4).

The conformational behavior of α-alanine in clusters, for the example of S-C_3_H_7_NO_2_@P(5,2), is presented in Scheme 13. In this case, the main minimum on the PES corresponds to the conformer **D**.

Comparison of clusters’ energy (Table 4) indicates that in a series of clusters with SWCNTs: (5,5), (5,1) and (5,2) their relative stability grows; the most stable are practically degenerated in energy endocomplexes S-C_3_H_7_NO_2_@P(5,2) and R-C_3_H_7_NO_2_@M(5,2) (Figure 19). Thus, the increase in chirality index m leads to a rise in the relative stability of the appropriate chiral clusters.

All results confirm the earlier conclusion [257] that the combination of S-enantiomer with P-SWCNT and R-enantiomer with M-SWCNT leads to the formation of most stable endocomplexes and creates a theoretical foundation for the recognition and effective separation of optical isomers.

## 8. Relative Stability of Cis and Trans Isomers inside Nanotubes. The “Trans-Effect”

It is well known that, in some cases, the cis isomer of compounds with a double bond is more stable than its trans form. This applies, for example, to 1,2-difluoroethene and difluorodiazene [261,262]. Such phenomenon is called the “cis effect”, and its probable causes have been discussed in the literature [263,264,265,266]. In particular, according to [266], the cis effect in the case of difluorodiazene is caused for a number of reasons. One of them is connected with a destabilization of the trans form because of the increased ionic character of the bonds linking the central to the peripheral atoms. This leads to the destabilizing reduction of the valence angles in both isomers, but in cis form this change is relaxed by Coulombic repulsion between two terminal atoms.

Using the PBE/3ζ approximation, it has been shown that free 1,2-difluoroethene and difluorodiazene demonstrate an advantage in energy in their cis form. However, investigation of the relative stability of both isomers in the cavity of SWCNT (4,4) demonstrates the action of the “trans-effect”: the predominance of the trans-configuration. The energy differences from the cis form reach two-digit numbers (Table 5) [267,268]. The same pattern is valid also for the cluster CHF=CHF@(6,6), although with less distinction in energy between both isomers.

The observed effect depends on the diameter of the nanotubes (Figure 20). It should be noted that there are several reasons for this complex phenomenon. On the one hand, the force field inside the nanotube significantly changes the structure of the guest species. In particular, the distance between fluorine atoms in cis-N_2_F_2_@(4,4) (2.402 Å) is shorter than for the free cis-difluorodiazene (2.452 Å), which enhances additional internal stress and leads to the destabilization of cis-configuration. On the other hand, in the case of cis-CHF=CHF@(4,4), the observed distance (3.084 Å) becomes greater than for the free cis-1,2-difluoroethene (2.793 Å). However, the cis isomer still remains unstable. Hence, it is also necessary to take into account the role of electronic factors: the noticeable change in Mulliken bond order (O_X=X_, Table 5) inside SWCNT (4,4) demonstrates a significant redistribution of electron density in the guest molecule, probably because of orbital interactions with the π-system of the nanotube.

Thus, SWCNTs are able to drastically change the relative stability of cis and trans isomers in their cavity for several reasons.

## 9. Conclusions

Promising opportunities of different nanostructured objects are largely connected with the formation of endohedral complexes of fullerenes and nanotubes. Unique conditions in their cavities can be viewed as the action of a solvent with specific properties. As a result, a combination of electronic and steric requirements, that may be called a force field, inside fullerenes and nanotubes significantly changes the stereochemical properties of even relatively simple molecules in comparison with their free state or with the action of usual solvents. Principal differences are mainly the following:The preferred conformation of ethane and its analogs inside nanotubes of small diameter is not the staggered, but the eclipsed form. The conformational behavior of ethane-like molecules inside fullerenes is less clear and depends on the size and chemical composition of the nanoobject.Hydroxyborane and diborane in endohedral clusters of nanotubes are characterized by reducing the barriers of internal rotation about the B-O and B-B bonds; in the case of dialane, the planar form—the transition state for the free molecule—becomes the minimum on the potential energy surface. A similar change in the nature of the preferred conformation is observed for hydrazine, methanol, methanethiol and dimethyl ether, together with cyclic molecules: cyclohexane, 1,3-dioxa-2-silacyclohexane and hexahydropirimidine-2-one inside nanotubes. In the case of 1,3-dioxane inside SWCNT and 1,3,2-dioxaborinane inside fullerene C_60_ conformational equilibrium is shifted to forms that can never be realized as a ground state for neither free molecule, nor for these species in any solvent.The barrier to pyramidal inversion of amines in endocomplexes varies depending on the nature of the amine and the size of the nanocavity.Chiral nanotubes are able to recognize molecules of enantiomers owing to increased affinity of *P*-SWCNT for the *S*-isomer and *M*-SWCNT for the *R*-form.SWCNTs of small diameter are able to drastically change the relative stability of cis and trans isomers in their cavity.

Everything mentioned above indicates fundamental changes in the structural properties of the guest species, and constitutes the ground for an in-depth systematic stereochemical investigation of a large variety of compounds inside nanotubes and fullerenes.

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
