# Peer review of "Stereochemistry of Simple Molecules inside Nanotubes and Fullerenes: Unusual Behavior of Usual Systems"

_molecules, 2020, doi:10.3390/molecules25102437_

Round 1

Reviewer 1 Report

The manuscript “Stereochemistry of Simple Molecules Inside Nanotubes and Fullerenes: Unusual Behavior of Usual Systems” reviews the contributions of different works towards the elucidation of the conformation states of different common molecules when they are confined in the nanocavities of such nanomaterials forming endohedral complexes. Most of the works cited refer to the same PBE model for the calculations from nearly the same authors. Many of them are published in journals which are not of general use. Therefore this work can serve to draw near all the work compiled in such journals. It seems clear that the force field of the nanocavity influences dramatically the stereochemical characteristics of the guest molecule. Moreover, the size and chiral characteristics of the host system also condition the stability of R- and S- enantiomers. The influence of the host system in the energy of pyramidal inversion of nitrogen atoms in amine derivatives is also analyzed. The work is quite systematic so it can help as a reference work for those working in the formation of endohedral complexes inside carbon nanomaterials. The conclusions summarize quite briefly the most important general aspects derived from the works mentioned throughout the text. Therefore, I consider the work interesting enough to deserve publication in Molecules. I have only detected some minor error, most of them of typographic character. Please find them below:

- Line 194: Typo in nanotube’s diameter

- Line 259: Section 3.2 refers to Diborane(4) but uses the generic term Diborane which is commonly used for Diborane(6). This is confusing and should be specified. Same is valid for alane AlH4.

- Line 271: Typo in main minimun.

- Line 273: Typo in growth.

- Line 304: Typo in All.

Reviewer 2 Report

In the Manuscript, the Author presents extensive review of endohedral host-guest complexes based on carbon fullerenes or nanotubes. Such complexes attract significant interest and are regarded as a building blocks for novel materials. The presented review provides systematization of the results of recent investigations and is potentially highly citated. I recommend it for publication in Molecules after some revisions:

1) It is not clear how the regarded inner molecules were chosen. Some of that are small hydrocarbons, but others are non-carbon compounds. What was the reason to select C-O, C-S, B-B, B-O, B-N, N-N, Al-Al, Si-Si and Ge-Ge bonds (These bonds are more interest than others? Two first lines of periodic Table? Other molecules were not investigated?). Please add some clarifying details.

2) It should be mentioned that fullerenes and nanotubes stabilize high-strained molecules that possess low stability without carbon skeleton (nitrogen cubane inside the fullerene cage; another small nitrogen non-molecular clusters into nanotubes; tetrahedrane C4H4 in C60; etc.).

3) The Author declare that PBE and wB97XD methods are the most suitable for considered complexes. This statement is controversial; the Authors of mentioned references were not compare these methods with others. For small molecules, B3LYP works better than PBE. The main advantage of wB97XD method is implementation of non-covalent interaction, which is very important in host-guest compounds. However, other methods also can account non-covalent attraction via dispersion corrections.

Reviewer 3 Report

I suppose that some attention should be devoted to behavior of different molecules inside the cavity of icosahedral fullerenes (such as C60, C240, C540). It's known that energy wells in these fullerene are determined and can be predicted by its topology (doi: 10.1002/wcms.1207, https://doi.org/10.1016/S0009-2614(97)01121-4). Earlier the behavior of water clusters (https://doi.org/10.1016/j.chemphys.2011.05.025) and other fullerens (DOI: 10.1002/jcc.23620) inside icosahedral fullerenes was studied.

At the page 2 lines 54-55 authors mention the possibility of inside molecule motion in the cage of carbon nanoutbe. I believe that readers will be interested if the manuscript will contains more examples of such systems beside the link [27]. Here are some modern papers with interesting mechanism of functioning that can be cyted: https://doi.org/10.1063/1.5083846,  DOI: 10.1115/ICNMM2017-5563, https://doi.org/10.1016/j.taml.2018.04.007, https://doi.org/10.1021/acs.jctc.0c00009
